# Revisiting Porcine Circovirus Infection: Recent Insights and Its Significance in the Piggery Sector

**DOI:** 10.3390/vaccines11081308

**Published:** 2023-07-31

**Authors:** Hemanta Kumar Maity, Kartik Samanta, Rajib Deb, Vivek Kumar Gupta

**Affiliations:** 1Department of Avian Science, Faculty of Veterinary & Animal Science, West Bengal University of Animal & Fishery Sciences, Kolkata 700037, West Bengal, India; 2ICAR-National Research Center on Pig, Rani, Guwahati 781131, Assam, India

**Keywords:** porcine circovirus, vaccine, prevention

## Abstract

Porcine circovirus (PCV), a member of the *Circoviridae* family within the genus *Circovirus*, poses a significant economic risk to the global swine industry. PCV2, which has nine identified genotypes (a–i), has emerged as the predominant genotype worldwide, particularly PCV2d. PCV2 has been commonly found in both domestic pigs and wild boars, and sporadically in non-porcine animals. The virus spreads among swine populations through horizontal and vertical transmission routes. Despite the availability of commercial vaccines for controlling porcine circovirus infections and associated diseases, the continuous genotypic shifts from a to b, and subsequently from b to d, have maintained PCV2 as a significant pathogen with substantial economic implications. This review aims to provide an updated understanding of the biology, genetic variation, distribution, and preventive strategies concerning porcine circoviruses and their associated diseases in swine.

## 1. Introduction

Porcine circovirus (PCV), a ubiquitous viral pathogen of pigs, belongs to the genus *Circovirus* and family *Circoviridae*, and is responsible for porcine circovirus virus-associated disease (PCVAD), which includes a number of diseases such as post-weaning multisystemic wasting syndrome (PMWS), porcine dermatitis and nephropathy syndrome (PDNS), granulomatous enteritis, porcine respiratory disease complex, reproductive failure, and acute pulmonary edema [1,2]. PCV was first discovered as a picornavirus-like contamination in a permanent porcine kidney cell culture (PK-15 ATCC-CCL31) with no cytopathic effect and proposed to be an RNA virus [3]. Later studies revealed that the virus is, in fact, a non-enveloped, small circular DNA virus with an icosahedral capsid structure [4]. Subsequently, a new type of PCV infection emerged in Canada [5,6], North America, and Europe [7,8]. Sequence analyses of the newly emerged PCV demonstrated significant differences when compared to the previously known noncytopathic PCV [9]. To differentiate the newly found pathogenic virus, it was designated Porcine circovirus type 2 (PCV2), while the non-pathogenic virus was named Porcine circovirus type 1 (PCV1) [10]. Despite their distinct pathogenic properties, both PCV1 and PCV2 viruses seem to have a common origin [11]. At present, the genus Circovirus consists of four species, PCV type 1 (PCV1), PCV type 2 (PCV2), PCV type 3 (PCV3), and PCV type 4 (PCV4) [12]. PCVs are widespread in both domestic pigs and wild boars [13,14]. Although PCV is a primary pathogen of swine sources, it has been occasionally reported in non-porcine animals, including ruminants (such as cattle, goat, and roe deer), rodents (such as house mice and black rat), and carnivores (such as dogs, minks, foxes, and raccoon dogs). Additionally, PCV has been detected in insects (such as flies, mosquitoes, and ticks), shellfishes, biological products (such as vaccines and pig-derived commercial pepsin products), and even in environmental samples such as bio-aerosol and water [15,16]. In this review, we discuss the origin, distribution, genetic diversity, transmission, and life cycle of porcine circoviruses. Furthermore, diagnostic approaches for PCV infections and controlling strategies are also discussed.

### 1.1. Genomic Organization

PCV is a small, non-enveloped, icosahedral, single-stranded circular DNA virus (Figure 1) with a diameter ranging from 13 to 25 nm [17]. The genome sizes of PCV1, PCV2, PCV3, and PCV4 are 1758–1760 nt, 1767–1777 nt, 1999–2001 nt, and 177 nt, respectively [17,18,19,20].

The genomic DNA sequence homology between PCV1 and PCV2 ranges from 68% to 76% [21,22]. Both PCV1 and PCV2 consist of 11 predicted open reading frames (ORFs). Among these ORFs, ORF1, ORF5, ORF7, and ORF10 are located on a positive strand and transcribed in a clockwise direction. To the contrary, ORF2, ORF3, ORF4, ORF6, ORF8, ORF9, and ORF11 are located on a negative strand and are transcribed in a counterclockwise direction [21]. A summary of major ORFs of different PCVs is presented in Table 1 [17,18,19,20,21,23,24,25] and genomic similarities among different PCV species are illustrated in Table 2 [12,20,21,22,23,24,26,27].

There are two major ORFs in PCV, namely ORF1 and ORF2, which are oriented in the antisense direction [28]. The ORF1s of PCV 1 and PCV2 share a substantially high degree of similarity, with 83% of their nucleotide sequences and 86% of their amino acid sequences being identical in ORF1 regions [22,29]. The primary function of ORF1 is to encode two viral replication initiator proteins, namely Rep and Rep’. Rep is encoded by the entire length of the ORF1 transcript, whereas Rep’, a truncated version of Rep, is produced by alternative splicing of ORF1 [25]. In PCV1, Rep and Rep’ consist of 312 amino acids (aa) and 168 amino acids (aa), respectively, while in PCV2, Rep and Rep’ are composed of 314 aa and 297 aa, respectively, [17,21]. The ORF2 of PCV1 and PCV2 showed 67% and 65% similarities at nucleotide sequence and amino acid sequence levels, respectively [22,29]. The ORF2 encodes the viral capsid (Cap) protein of 230–233 aa in PCV1 and 233–236 aa in PCV2 [17,21,30,31]. Capsid protein (Cap) is a structural component of PCV [32] and is highly immunogenic as it contains antigenic epitopes [33].

ORF3 is the most variable open reading frame between PCV1 and PCV2, with only 61.5% amino acid sequence identity [34]. It is embedded within ORF1 and oriented in the opposite direction [35]. ORF3 encodes a protein that is involved in viral pathogenesis through its apoptotic activity [36]. ORF4 is located within ORF3 and is oriented in the same direction [37]. The product of ORF4 is an anti-apoptotic protein that exerts its effect by reducing the level of ferritin heavy chain (FHC) protein in infected host cells [38]. ORF5 completely overlaps with ORF1 and is oriented in the same direction [39]. This ORF encodes a protein that promotes viral replication by inducing autophagy [40]. ORF6, which is located within ORF2 and oriented in the same direction, is found to regulate caspases and expression of several cytokines during PCV infection in cultured PK 15 cells [41].

The genome of PCV3 has at least three ORFs, ORF1, ORF2, and ORF3. ORF1 is oriented in the opposite direction to that of ORF2 and encodes a single 296–297 aa replicase (Rep) protein that shares 55% amino acid sequence similarity with PCV2 proteins [23]. ORF2 encodes a 214 aa capsid (Cap) protein that has 24% and 26–37% sequence similarity with PCV1 and PCV2 capsid proteins, respectively [23,24]. ORF3 encodes a 231 aa protein of unknown function [23].

The genome of PCV4 contains twelve predicted ORFs including two major ORFs, ORF1 and ORF2. ORF1 encodes a replicase protein of 296 aa that shares 48.1% and 47.2% amino acid sequence identity with PCV1 and PCV2, respectively. ORF2 encodes a capsid (Cap) protein of 228 aa that shows 43.1%, 45%, and 24.5% sequence identity with PCV1, PCV2, and PCV3, respectively [20].

### 1.2. Genotypes

According to a new genotyping methodology, PCV2 has eight distinct genotypes, PCV2a to PCV2h [42]. Recently, a new genotype designated as PCV2i has been also discovered in the USA [43]. The enormous genetic diversity of PCV2 is due to its higher evolutionary rate (~1.2 × 10^−3^ substitutions/site/year), like that of the RNA viruses [44]. Among nine genotypes, three major genotypes, namely PCV2a, PCV2b, and PCV2d, have been found throughout the world, while others have restricted distribution [42,43]. The worldwide occurrence of different genotypes of PCVs has been illustrated in (Figure 2).

Furthermore, PCV2a has five clusters (2A–2E), PCV2b has three clusters (1A-1C), and PCV2d has two sub-genotypes (2d-1, 2d-2) [11,45]. PCV2a was the dominant genotype until 2003, after which PCV2b emerged as the prevailing genotype worldwide [11].

The higher genetic variability within PCV2a compared to PCV2b suggests that PCV2a might be older genotype relative to PCV2b [46]. Phylogenetic analyses suggested that PCV2a and PCV2b likely originated from a common ancestor approximately 100 years ago. Since then, they have followed independent evolutionary pathways despite co-circulating in the same host species and geographic regions for years [44]. The genotypic switch from PCV2a to PCV2b occurred during 2004–2005, coinciding with the rapid emergence of PCV2-associated disease in Canada [47]. Since 2012, PCV2d become the predominant genotype and has been reported in many countries, including the USA, China, Thailand Korea, and Uruguay [48,49,50,51,52]. PCV-2c was initially identified from archived serum samples in Denmark [53]. Subsequently, it was also reported in Brazilian feral pigs and Chinese domestic pigs [54,55]. Another genotype, PCV-2e, first appeared in USA in 2006 [56], and subsequently reported in Mexico and China in 2015 [57,58]. A sixth genotype, PCV-2f, initially appeared in China before 1999 and then also identified in India, Croatia, and Indonesia between 2009 and 2014 [59]. Other two genotypes, PCV2g and PCV2h, were discovered when a new methodology for genotyping of PCV2 was proposed in 2018 [42]. All nine genotypes of PCV2 have been found in domestic pigs [43]. Apart from domestic pigs, the occurrence of PCV2 has also been reported in wild boar populations worldwide [60]. However, only seven of nine recognized genotypes (a, b, d, e, f, g, h) have been reported in wild boars [60,61,62,63]. Furthermore, the genotypes identified in wild boars exhibit a close genetic similarity to that found in domestic pigs, thus indicating the potential transmission of PCV2 from wild boars to domestic pigs [62]. Unlike PCV2d, which is the predominant genotype in domestic pigs around the world, PCV2b is the most common genotype in wild boars [61,62].

PCV3, which was initially reported in the US in 2016 [24], has also been detected in other countries, such as China [64]), South Korea [65], Poland [66], Italy [67], Brazil [68], and India [69]. Recently, a retrospective study revealed that PCV3 has been present in Sweden since 1993, long before its first report in the US [70]. On the basis of two amino acid mutations (A24V and R27K) in the Cap protein, PCV3 could be classified into three distinct clades, namely PCV3a, PCV3b, and PCV3c. Moreover, the PCV3a clade has three subclades: PCV3a1, PCV3a2, and PCVa3. This subdivision within clade PCV3a has been conducted based on observed phylogenetic relationships and other molecular characteristics in the Cap protein [71]. The occurrence of all three subtypes of PCV3 has been reported in domestic pigs. However, only two subtypes (3a and 3b) of PCV3 have been identified in wild boar populations, with PCV3b being the predominant subtype [13,72]. The phylogenetic studies have suggested that PCV3 likely originated approximately 50 years ago [73]. Notably, PCV3 shows a closely relationship with bat circovirus [74] and may have evolved from that virus before gradually adapting to infect both pigs and dogs [75].

PCV4 was first reported in 2019 in Hunan Province, China [20]. Subsequently, it was also detected in several other provinces of China, including Henan, Shanxi, Jiangsu, Anhui, and Guangxi [76], and in Korea as well [77]. However, a retrospective study has demonstrated the presence of PCV4 in archived samples dating back to 2012, suggesting that virus 4 has been circulating for at least 10 years in China [78].

On the basis of ORF1, ORF2, and complete genome sequences, all PCV4 strains can be categorized into two major genotypes (PCV4a and PCV4b) [79]. PCV4 is more closely related to mink circovirus (66.9%) as compared to PCV1 (50.3%), PCV2 (51.5%), or PCV3 (43.2%) [20]. The genetic and evolutionary relationship among different species of PCVs (PCV1, PCV2, PCV3, and PCV4) is illustrated as a phylogenetic tree based on the full open reading frame 2 (ORF2) gene sequences of PCVs retrieved from GenBank (Figure 3).

The phylogenetic tree was constructed using 13 complete ORF2 sequences of selective representatives of different PCVs (PCV1, PCV2, PCV3, and PCV4) obtained from GenBank. The tree was generated using the p-distance model-based neighbor-joining (NJ) method and bootstrapped at 1000 replications. Branches with a bootstrap vale of less than 70% were collapsed.

### 1.3. Clinical Signs and Pathological Lessions

Among four PCVs discovered so far, PCV1 is considered as non-pathogenic, while the other three PCVs, i.e., PCV2, PCV3, and PCV4, are known to be pathogenic. PCV1, PCV2, and PCV3 are distributed throughout the global pig population, while the distribution of PCV4 is yet to be known [80]. PCV2 is the main etiological agent of several diseases in pigs, collectively known as porcine circovirus-associated diseases (PCVADs) in North America [81] or porcine circovirus diseases (PCVDs) in Europe [82]. These diseases include PCV2-systemic disease (PCV2-SD) formerly known as postweaning multisystemic wasting syndrome (PMWS), PCV2-reproductive disease (PCV2-RD), PCV2-lung disease (PCV2-LD), PCV2-enteric disease (PCV2-ED), porcine dermatitis and nephropathy syndrome (PDNS), PCV2-subclinical infection (PCV2-SI) [83], and acute pulmonary edema [2]. PCV2-SD affects pigs between 8 and 16 weeks of age [84] and is clinically characterized by wasting, dyspnea, enlargement of lymph nodes, diarrhea, pallor of the skin, and jaundice [85]. Histopathological lesions include loss of B-cell follicles in lymph nodes, lymphocytic-histiocytic infiltration in lungs, liver, and kidney, depletion of mature lymphocytes in spleen, and atrophy of pancreatic acinar cells [85]. Morbidity and mortality in affected farms are 4–30% (occasionally 50–60%) and 4–20%, respectively [86]. The clinical signs of PCV2-RD include late-term abortions, stillbirths [87], and mummification [88]. PCV2-RD is rare under field conditions due to higher seroprevalence of PCV2 in adult pigs, and, as a result, breeding stocks do not show clinical symptoms [89]. Affected herds are often start-up herds, either naïve or with a higher number of gilts and PCV2 seronegative herds. The affected herds are frequently found to be either naïve or have a higher number of gilts, with the majority of them being PCV2 seronegative herds [90]. Pathologic lesions include hepatic chronic passive congestion, cardiac hypertrophy, cardiac fibrosing, and necrotizing myocarditis [91]. The main clinical signs of PCV2-LD include fever, cough, anorexia, and dyspnea [92], and are commonly seen in pigs of 14–20 weeks [93]. Histologically, PCV2-LD can be recognized by thickening of alveolar septa through mononuclear cell infiltration and bronchointerstitial pneumonia with peribronchial and peribronchiolar fibrosis [94]. Respiratory clinical signs of PCV2-LD may also be present in PCV2-SD and there is a potential diagnostic overlap between these two conditions [81]. PCV2-LD can be distinguished from PCV2-SD through histopathological findings and examination of lungs and lymphoid tissues (microscopic lesions are absent in lymphoid tissues in the case of PCV2-LD) [83].

PCV2-LD is equivalent to PCV2-associated porcine respiratory disease complex (PRDC) [95]. PCV2, along with several other pathogens such as porcine reproductive and respiratory syndrome virus (PRRSV), swine influenza virus (SIV) and Mycoplasma hyopneumoniae, play significant roles in the development of PRDC [84]. PRDC is a multi-factorial disease with a morbidity of 10–40% and a morality of 2–20% [96]. PCV2-ED is clinically characterized by diarrhea, granulomatous enteritis, and colitis [97]. Although PCV2-LD and PCV2-ED were previously considered as distinct clinical manifestations [83,97], recent studies indicate that PCV2-LD and PCV2-ED are negligible conditions and PCV2 mainly contribute to respiratory and enteric lesions in relation to the occurrence of PCV2-SD [95,98]. The primary clinical signs observed in PDNS-affected pigs are irregular macules and papules appearing on the perineal area of hindquarters, limbs, dependent parts of abdomen and thorax, and the ears. Skin lesions may occasionally merge, forming large irregular patches and plaques. Other clinical signs include anorexia, prostration, depression, stiff-gait, and moving reluctancy with no or mild pyrexia. PDNS affects mainly weaners and growing–finishing pigs, but also breeding adults sporadically [99]. Histologically, PDNS is characterized by enlarged, pale kidneys with cortical petechiae, acute glomerulonephritis, systemic necrotizing vasculitis, subcutaneous edema, and serous effusions in body cavities [100]. Morality rate ranges from 50% in younger pigs to 100% in pigs older than 3 months [83]. The predominant form of PCV2 infection is PCV2-SI, which is characterized by increase in average daily loss without clear clinical signs, minimal or no histopathological lesions in tissues, and presence of a low amount of PCV2 in tissues [83]. PCV2 is also associated with acute pulmonary edema (APE) in nursery and younger finisher pigs, characterized by rapid onset of respiratory distress, leading to immediate death. The development of pulmonary edema in APE arises from the loss of integrity of the blood vessel wall, caused by endothelial cell damage and the release of cytokine by monocytes, leading to the outflow of vascular contents into the interstitium [2]. Both domestic pigs and wild boars are susceptible to PCV2 infection [62]. Several studies have demonstrated that infection of PCV2 in wild boar also causes development of PCV-SD, which is manifested by weight loss, dyspnea, wasting, etc. Histopathologic lesions in PMWS-affected wild boars are identical to those of domestic pigs [101,102,103].

PCV3 has been detected in pigs showing the symptoms of systemic disease, reproductive disease [104], PDNS [23,105]), respiratory disease [106], GI disorder [107], PFTS [108], and cardiac and multisystemic inflammation [24]. The major clinical signs include mummified and stillborn fetuses [104,109], weight loss, dyspnea, rectal prolapse [24], abdominal breathing, diarrhea [107], anorexia, fever, icterus [110], congenital tremor [111], and wasting [112]. Pathological lesions that are associated with PCV3 infection include lymphoid depletion in the spleen and lymph nodes [113]; periarteritis and arteritis in the heart, kidney, spleen, lung, and/or stomach [112,114]; bronchointerstitial pneumonia in the lung [23]; epicarditis, myocarditis, and endocarditis in the heart [104,113,115]; atrophic, shortened villi and decreased depth of crypt in the intestine [64]; and mild lymphoplasmacytic meningoencephalitis, nephritis, periportal hepatitis, and rhinitis [112]. Recently, a study also demonstrated the potential role of PCV3 in subclinical infection, which is characterized by lack of evident clinical signs but presence of detectable prolonged viremia, viral replication in tissues, and multisystemic inflammation [116]. PCV3 can infect animals of different age groups and production phases, being found in fetuses, nursery pigs, fattening pigs, stillborn, and sows [17,117]. PCV3 has been detected not only in pigs with signs and symptoms of different clinical diseases [23,24], but also in healthy animals [104]. Moreover, co-infection of PCV3 with other viruses such as PCV2 [118], PRRSV [106], PPV [108], CSFV [118], and TTSuVs [119] has been reported. Thus, the detection of PCV3 in samples alone does not guarantee its association with disease casualty [114]. Under experimental conditions, an infectious PCV3 DNA clone was able to induce development of PDNS-like signs in specific-pathogen-free (SPF) piglets [120]. However, in another study, no clinical disease was developed in caesarean-derived, colostrum-deprived (CD/CD) pigs inoculated with PCV3. However, four out of eight PCV3-infected pigs showed histological lesions such as lymphoplasmacytic myocarditis and perivasculitis, which are consistent with multisystemic inflammation [109]. Similar results were found in PCV3-inoculated CD/CD pigs where no significant clinical signs were observed, except histological lesions that resembled multisystemic inflammation. Thus, the pathobiology of PCV3 is complex and of multifactorial nature [116]. Further investigation is needed to reveal the pathogenic efficacy of PCV3 along with the role of other co-infecting pathogens. PCV3 infects both domestic pigs as well as wild boars. However, there is still no evidence regarding the association of PCV3 infection and development of clinical disease in wild boar [121].

Infection with PCV4 shows the symptoms of respiratory disease, enteric disease, PDNS, and PMWS [20,122]. The virus has been identified in pigs of all age groups, including aborted fetuses, suckling piglets, weaners, growers, finishers, and sows [77]. Major clinical manifestations are diarrhea, pulmonary oedema, skin lesions, neurological symptoms, and aborted and stillborn fetuses [20,122]. Diseased pigs with clinical symptoms of PCV4 infection were also found to be simultaneously infected with other PCVs such as PCV2 and PCV3 [122]. Moreover, PCV4 was also isolated from clinically healthy pigs [77], as well as from pigs that were simultaneously co-infected with PCV2 and PCV3 [123]. Therefore, more research is essential to determine the pathogenic role of PCV4, along with its clinical significance and/or contribution of co-infecting pathogens.

## 2. Factors Associated with Clinical Manifestation of PCV-Associated Disease (PCVAD)

### 2.1. Virus-Dependent Factors

Although PCV2 is the primary pathogen of PCVAD, it is alone not sufficient to cause clinical disease. Several other infectious co-factors and non-infectious conditions are crucial for the clinical expression of PCV2 infection [124]. PCV2 infection may downregulate the host immune system and thus pave the way for the infection of other pathogens. A number of microbial pathogens have been found to co-infect and increase the severity of PCVAD in pigs. Some of these are *Porcine parvovirus* [125], *Porcine reproductive* and *respiratory syndrome virus* [126], *Swine Hepatitis E virus* [127], *Swine influenza virus*, *Mycoplasma hyopneumoniae* [128], *E. coli* [129], and *Salmonella* spp. [130]. A retrospective study of PCV2-infected swine samples has shown that PRRSV was the most frequent coinfecting pathogen (51.9%), followed by *M. hyopneumoniae* (35.5%), bacterial septicemia pathogens (14%), bacterial pneumonia pathogens (7.6%), and SIV (5.4%) [131]. Moreover, coexistence of different genotypes of PCV2 in the same pig has been reported from different countries, including China and the USA [132,133]. Such coexistence may contribute to the development of more severe clinical symptoms in pigs [132]. Several studies have also reported the co-infection of different PCVs, such as PCV2 and PCV3, as well as PCV1 and PCV3 [134,135]. Different PCV2 genotypes have been associated with varying degrees of clinical manifestations and severity of PCVAD. For example, in two studies, PCV2b was found to be more pathogenic and commonly associated with the occurrence of PCV-SD as compared to PCV2a [46,136]. However, other experiments did not reveal a significant difference between PCV2a and PCV2b with respect to pathogenicity. Furthermore, different isolates within the same cluster exhibited a significant difference in virulence [137]. Evidence has suggested that virulence is a function of the specific PCV2 isolate, regardless of the genotype [138]. A study investigating the relative virulence of three major genotypes of PCV2 (2a, 2b, and 2d) revealed that PCV2d is more virulent than PCV2a or PCV2b [139]. However, other studies have shown that there is no significant difference among PCV2a, PCV2b, and PCV2d with respect to virulence [140,141]. Nonetheless, the involvement of a co-infecting pathogen might play a role in causing a difference in virulence among different genotypes. A study by Suh et al. (2021) proved that PCV2a, PCV2b, and PCV2d show significant differences in virulence when co-infected with PRRSV-2, whereas no significant virulence difference was seen when each genotype infected solely. PCV2d was more virulent than PCV2a and PCV2b in a dual-infection model [142]. Another experiment involving *M. hyopneumoniae* as a co-infecting agent revealed the same results [143]. Such co-infectious agents might potentiate the replication of PCV2d, as reflected in greater severity in lymphoid lesions [142,143]. Therefore, further studies are essential to explore the relation of the genotype, isolate, or co-infecting agents with the pathogenic expression of PCV2 infection. Capsid protein, the product of ORF2, is a determining factor of antigenicity and virulence in PCV2 [144]. Mutation in the PCV2 cap gene might play a role in the alteration of viral pathogenicity. Under experimental conditions, it was observed that PCV2 could lose its pathogenicity after 120 serial passages in PK-15 cells due to two mutations (P110A and R191S) in the cap gene [145]. Mutation in the cap gene, even involving a single amino acid, may alter the virulence and pathogenicity of PCV2 [146]. Several studies have shown that a genotypic shift has coincided with more severe outbreaks of PCVAD in different countries [47,53,147,148].

### 2.2. Host-Dependent Factors

While PCV2 infections are widespread, the prevalence of PCVAD is relatively lower [83]. Consequently, not all pigs with PCV2 infection exhibit the clinical signs of PCVAD. Only a small percentage of PCV2-infected pigs develop clinical disease. Under field conditions, susceptibility to develop clinical disease varies among different genetic breeds of pigs [35]. For example, an experimental study demonstrated that among three pig breeds—Duroc, Landrace, and Large white pigs—only Landrace pigs develop clinical symptoms of PCV-SD, despite all three breeds being equally susceptible to PCV2 infection [149]. Likewise, in other experiments, YL (Yorkshire × Landrace) pigs displayed severe clinical symptoms, whereas LW (Laiwu) pigs exhibited only minimal clinical lesions [150]. The results of different experiments indicate that genetic differences are the underlying cause of varying susceptibility or resistance to PCV2 infection among different pig breeds [151].

### 2.3. Effect of Immunomodulation

The level of maternal antibodies (MDAs) could play a critical role in determining the host’s resistance to PCV2 infection. A higher level of maternal antibodies results in increased resistance to viral infection in piglets [152]. The levels of maternal antibodies in a host pig are inversely correlated with its age, meaning that as pigs grow older, the level of maternal antibodies decreases. Consequently, the older pigs have a lower level of maternal antibodies, leading to a significantly higher level of PCV2 viremia as compared to younger pigs [153]. Thus, maternal antibodies provide protection against PCV2 in a titer-dependent manner, i.e., with higher the level of maternal antibodies, the greater the protection the pigs will have [152]. PCV2 infection may also modulate the activity of the host cellular immune system, which, in turn, is associated with the pathogenesis of PCVAD. PCV2 blocks the ability of natural interferon-producing cells (NIPCs) to produce IFN-α, which ultimately favors PCV2 survival and secondary infections [154]. Furthermore, PCV2 upregulates IL-10 secretion by monocytic cells, which subsequently inhibits IFN-γ, IFN-α, and IL-12 secretion by PBMCs, thus causing suppression of Th1 responses and immune responses and favoring co-infections [155].

### 2.4. Management Factors

Epidemiological studies have shown that on-farm management factors, housing conditions, vaccination schedules, biosecurity, hygiene, and husbandry practices are strongly associated with the development of PCVAD [156]. Large pens in weaning facilities, proximity to other pig firms, vaccination against PRRSV, short empty periods in nursery and farrowing sectors, early weaning, use of farm boars, etc., would act as risk factors, whereas shower facilities and vaccination of sows against *E. coli* and atrophic rhinitis would act as protective factors [156,157]. The combinatorial effect of these factors might regulate the duration as well as the clinical manifestation of PCV2 infection. Environmental stressors such as change in temperature, mixing, noise, and shipping may suppress the host immune system, and thus increase susceptibility to PCVAD. Such stressors are alone sufficient to develop clinical signs in PCV2-infected pigs [158].

### 2.5. Vaccine-Related Factors

Vaccines utilized for co-infecting pathogens or adjuvants utilized along with those vaccines may also affect the outcome of PCV-2 infection. In an experimental study, gnotobiotic piglets vaccinated with a commercial *Mycoplasma hyopneumoniae* (*M. hyopneumoniae*) bacterin containing a mineral oil adjuvant exhibited PMWS after vaccination. Clinical signs were developed only in forty percent (40%) of vaccinated piglets [159]. Similarly, prenatally PCV2-infected pigs developed PMWS postnatally only when they were injected with keyhole limpet hemocyanin emulsified in incomplete Freund’s adjuvant (KLH/ICFA) [160]. The age of vaccinated pigs might also affect the severity of PCV2-associated lesions [161]. For example, under experimental conditions, three groups of pigs (AGE2, AGE7, and AGE12) were vaccinated with a commercial *Mycoplasma hyopneumoniae* vaccine at 1 week of age and challenged with PCV2 at 2, 7, and 12 weeks of age; the AGE12 group of pigs had higher levels of viremia compared to nonvaccinated pigs. However, this effect was not observed in younger pigs (AGE2 and AGE7) [153]. Although there are several pieces of evidence for the induction of PCV2 lesions by adjuvants, not all adjuvants can trigger the higher incidence of PMWS development [162]. Furthermore, not all adjuvants equally contribute to the occurrences of PCV2-associated lesions. The severity of lymphoid depletion was greater in the groups treated with *M. hyopneumoniae* vaccine along with oil-in-water adjuvant, compared to the groups treated with the same vaccine but in conjunction with different adjuvants, i.e., aqueous-carbopol and aluminum hydroxide adjuvants [163]. Many studies, both in experimental models and field conditions, have indicated that immunostimulation by vaccines enhances PCV2 replication and chances of clinical manifestation of PMWS [164,165,166]. However, other studies did not support the immunostimulation as a critical factor of development of PMWS [167,168].

### 2.6. Tramission

Both horizontal (among pigs of the same generation) and vertical (mother to offspring) modes of PCV2 transmissions have been reported. However, the horizontal mode of transmission is very frequent. A recent study indicated that direct contact between pigs is the most efficient means of PCV2 transmission [169]. However, indirect transmission through contaminated vectors or fomites may also occur [170]. The experimental evidence indicates the direct transmission of PCV2 from previously inoculated or naturally infected pigs to naïve or SPF pigs, leading to seroconversion and development of PCVD in recipient pigs [171,172]. Additionally, there is also the possibility of indirect airborne transmission of PCV2 at detectable concentrations [173]. Apart from the oronasal route, which is the most likely means of horizontal transmission [82], other potential routes of PCV2 transmission are the nose-to-nose route [174] and fecal–oral route [175]. Surprisingly, not only the virus but also PMWS showed the potentiality of transmission from PMWS-affected pigs to healthy, unaffected pigs through direct contact or indirect airborne contact [176,177]. The virus might be present in nasal secretion, trachea-bronchial secretion, blood, urine, faces, oral fluid, milk, and semen [178,179,180,181]. PCV2 could be shed by both diseased and clinically healthy pigs, but diseased pigs shed more virus than sub-clinically infected and clinically healthy pigs [178]. Several experimental observations confirmed that PCV2 could be shed in milk of infected sows and could be transmitted to offspring by an oral route [180,182,183]. Even vaccination could not prevent the presence of PCV2 in colostrum and milk but can only reduce its shedding. The shedding of infectious PCV2 at higher frequencies in colostrum and milk has been reported under experimental conditions, even in the presence of high neutralizing antibody (NA) titers [184]. PCV2 may also be transmitted from infected boars to sows and piglets via semen and could pose a potential risk to the herd [185]. Both PCV2a and PCV2b shed in semen were shown to be infectious in a swine bioassay model. However, low doses (105.6–105.8 genomic copies/mL) of PCV2 in semen did not cause reproductive failure, seroconversion, or viremia in naïve gilts and their offspring after artificial insemination. Thus, the amount of PCV2 in semen might be a critical factor in determination of PCV2 transmission via semen among swine herds [186]. The maternal antibodies play a protective role for the fetus by reducing the number of viruses that can cross the placenta. It was experimentally observed that the low level of maternal antibodies might increase the probability of fetal infection through the intrauterine route in PCV2-seropositive gilts when inseminated with PCV2b-spiked semen [187]. Trans-placental spread of PCV2 has been reported under both experimental and field conditions, which causes reproductive failure in pregnant sows [188], stillbirths [189], and fetal death with mummification [89]. Vaccination of dams, although inducing the production of neutralizing antibodies, cannot prevent vertical transmission of PCV2 [190]. PCV2 is transmitted primarily from infected pigs to healthy pigs. Additionally, PCV2 can also be transmitted from wild boar to domestic pigs. Wild boar could act as a reservoir as well as a vector for PCV2 [191,192]. Numerous studies from various countries have demonstrated that the genotypes of PCV2 found in wild boars closely resemble those in domestic pigs. This genetic similarity suggests an ecological interaction between wild boar and domestic pigs, which contributes to the transmission of PCV2 [62]. The detection of a viral genome in invertebrate animals such as housefly (*Musca domestica*), stable fly (*Stomoxys calcitrans*) and mosquito (*Culex* sp.) suggested that these insects may act as a mechanical vector of PCV2 [193,194,195]. The house fly has on-farm potentiality of carrying and transmitting PCV2b, as its life cycle stages are closely associated with pigs and their habitat [194]. PCV2 transmitted primarily from infected pigs to healthy pigs. However, interspecies transmission of PCV2 from pigs to other animals and vice versa has also been reported. A number of experimental findings indicated that such cross-species infection may occur from pig to buffalos [196], pig to fox [197], pig to mink [198], pig to rat [199], and pig to raccoon dog [200].

Like PCV2, PCV3 can be transmitted both horizontally and vertically [201]. The primary mode of viral transmission is through the horizontal route, involving direct contact [202]. PCV3 has been detected in various parenchymal tissues, blood, oral fluids, nasal swabs, feces, and semen [17,203]. PCV3 can also be found in colostrum, indicating the potential for vertical transmission from mother to piglets and its direct association with reproductive failure [203,204]. The presence of the PCV3 genome in mummies and stillborn fetuses highlights the ability of PCV3 to cause intrauterine infections [205]. Wild boar is a potential reservoir for the circulation of PCV3 [72,121]. The increasing population of wild boar, leading to more interactions with domestic pigs, could facilitate the transmission of PCV3 from wild boar to domestic pigs [72]. Furthermore, interspecies transmission of PCV3 from pigs to other animals might also be possible, as revealed by different studies [206,207].

As PCV4 is a very recently identified PCV, details about its transmission are not available. The virus has been detected in tissue specimens of respiratory, renal, digestive, circulatory, and lymphatic systems, as well as from aborted fetuses [77,78,123]. Because of its wide tissue distribution and tissue tropism, PCV4 has the potential for multi-route transmission, including transplacental infection [77].

### 2.7. Diagnosis

The diagnosis of PCVAD relies on three criteria: evaluating clinical signs, examining microscopic lesions in affected organs such as lymphoid tissue, lungs, liver, heart, kidney, and intestine, and detecting the PCV2 antigen or its DNA in lesions [86]. When a farm is affected by PCVAD, the following clinical signs are commonly observed: wasting (98.1%), diarrhea (77.2%), dyspnea (75.1%), lymphadenopathy (44.8%), central neurologic signs (39.6%), jaundice (37.1%), inappetence (90.4%), and death (96.8%) [24]. Other clinical signs on an affected farm may include elevated rates of abortion, stillbirth, and fetal mummification [208]. Microscopic lesions in lymphatic tissues may be manifested as lymphatic depletion, histiocytic infiltration, inclusion bodies, and giant cells [209]. Other microscopic lesions associated with PCVAD are necrotizing vasculitis and fibrinonecrotic glomerulonephritis in the dermis, kidney, spleen, and liver; bronchointerstitial pneumonia with peribronchial and peribronchiolar fibrosis in the lungs; myocardial degeneration or necrosis with oedema; and mild fibrosis, diffused moderate infiltration of lymphocytes, and macrophages [1].

Herd diagnosis as compared to diagnosis of a single pig is critical for the prevention and control of PCVAD. Such herd diagnosis can be conducted on the basis of two criteria: (i) a significant increase in postweaning mortality associated with clinical signs compatible with PMWS; and (ii) individual diagnosis of PMWS in at least 1 out of 3–5 necropsied pigs [210]. Furthermore, elimination of any other potential cause of higher mortality is also required. However, a clear case definition for PCVADs other than PMWS is yet to be proposed [83].

Currently, common techniques used for the detection of PCV2 and PCV3 antigens or nucleic acids include immunohistochemistry (IHC), in situ hybridization (ISH), and PCR [81,211,212]. Both IHC and ISH enable localization of PCV in infected tissues or cells, providing cellular details and histological architecture, thus allowing simultaneous observation of a number of infected cells and characteristic histopathological lesions in the same section [213]. There are varying opinions regarding the sensitivity and specificity of these two techniques. While some studies suggest IHC is more sensitive than ISH [214,215], others indicate ISH is more sensitive than IHC [213,216,217]. Additionally, some studies found both IHC and ISH to have equal sensitivity [218]. Similarly, some studies revealed that IHC is less specific than ISH [216], while others consider IHC to be more specific than ISH [215]. Furthermore, IHC has an advantage in terms of easier interpretation of the results due to better image quality after staining [215]. IHC uses monoclonal or polyclonal antibodies to detect the location of PCV2 and PCV3 antigens in infected formalin-fixed, paraffin-embedded tissue sections [81,219]. However, formalin used for tissue fixation can cause crosslinking of antigens, epitope masking, and decreased immunoreactivity [220]. Prolonged formalin fixation may also lead to decreased antigenicity limiting the use of formalin-fixed tissues for diagnostic IHC [221]. Hence, not all monoclonal and polyclonal antibodies are suitable for PCV antigen detection by IHC. In contrast, ISH is less susceptible to structural alternation caused by formalin fixation [222]. Moreover, ISH is useful for differentiating between different genotypes of PCV such as PCV2a and PCV2b, which is not possible by IHC [213]. However, use of ISH is limited as it is more technically complex and expensive than IHC [223].

PCR can also be used for the detection of PCV2 in infected tissues. PCR is fast, has higher sensitivity as compared to IHC and ISH, and can even be performed on live samples [211]. Conventional PCR is time-consuming and prone to sample contamination, which increases the chance of false positive results [224]. On the other hand, real-time PCR (qPCR) allows detection of PCV2 and PCV3 specifically, rapidly, and quantitatively without any false positive results or pollution [225,226]. Different types of qPCR, such as SYBR Green I-based qPCR and TaqMan-based qPCR, have been used for the detection of PCV2 and PCV3 [224,225,226,227]. The TaqMan-based quadruplex real-time PCR was able to differentiate four species of PCV strains in clinical samples, rapidly and simultaneously [228]. Recently, multiplex real-time PCR has been developed, which is rapid, sensitive, efficient, and highly specific, allowing the detection and discrimination of different PCVs from clinical samples and even comparison of different genotypes of a particular species without requiring any specialized laboratory equipment [229,230]. Droplet digital PCR (ddPCR) is another novel PCR technology used for the detection of PCV2 and PCV3 with greater analytical sensitivity than TaqMan real-time PCR [231,232]. Nanoparticle-based PCR could also be used for rapid diagnosis of single or concurrent PCV2 and PCV3 infections, and is an extremely sensitive and accurate method in contrast to other PCR methods [233]. Despite its advantage, PCR could not be used as the sole diagnostic technique for the confirmation of a clinical disease such as PMWS because PCV2 and PCV3 are ubiquitously present in swine, regardless of disease status. Thus, both PCV2 and PCV3 may also be present in clinically healthy pigs. PCR may be considered an alternative tool for the diagnosis of clinical disease only when it is interpreted in conjugation with characteristic histopathological lesions in tissues [216,234].

Serological techniques such as indirect immunofluorescence assay (IIFA) [235,236] and enzyme-linked immunosorbent assay (ELISA) [235,237] have been developed and utilized for the detection of anti-PCV antibodies in pigs. However, such tests cannot be used for the diagnosis of a clinical disease, as antibodies against PCV2 or PCV3 may also be present in clinically healthy pigs. Other techniques that could be used to detect and observe PCV2 and PCV3 include loop-mediated isothermal amplification (LAMP) [238,239], recombinase polymerase amplification assay (RPA) [240,241], and electron microscopy [242,243].

The detection of newly identified PCV4 has also been undertaken by conventional PCR [122], SYBR Green I-based real-time PCR [123], indirect ELISA [244], and LAMP [245], in the same way as for other PCVs. Recently, multienzyme isothermal rapid amplification (MIRA) assay [246] and CRISPR Cas13a based lateral flow strip (LFD) assay [247] have been developed for rapid detection of PCV4, and showed promising results in terms of specificity and sensitivity for the detection of the virus in clinical samples.

### 2.8. Life Cycle

The complete life cycle of PCV is yet to be explored. Due to their small genome size and limited coding capacity, the life cycle of PCV relies on host factors [248]. The successful attachment of the virus to its host cell is the first step in the infection cycle of any virus. Viruses typically use glycosaminoglycans (GAGs) for their attachment to the host cell surface (Figure 4). PCV2 specifically uses heparan sulfate (HS) and chondroitin sulfate B (CS-B) as its attachment receptors [249]. It was experimentally proved that PCV2 enters different cells through different routes. For example, PCV2 enters porcine monocytic 3D4/31 cells, primary porcine monocytes, and dendritic cells via clathrin-mediated endocytosis [154,250,251], whereas the same virus enters porcine epithelial PK-15 cells via the actin- and Rho GTPase-dependent pathways [252]. Sometimes, the virus enters the same host cell using different routes. For example, PCV2 enters porcine L-lymphoblasts via both clathrin-mediated endocytosis and macropinocytosis [253]. The routes of entry into the same host cell also determine the productivity of an infection. The actin- and small-GTPase-dependent internalization leads to productive infection in epithelial cells, whereas clathrin-mediated internalization does not lead to full infection due to incomplete replication [252]. Actin polymerization is critical for all types of PCV2 internalization and infection [251,252,253]. Immediately after internalization, the virus is transported through the vesicular trafficking pathway (early endosome–late endosome-endolysosome). Cytoplasmic dynein is a minus-end directed motor protein that uses the microtubular track for the cytoplasmic transport of cargo molecules including virus particles [254]. In the case of PCV2, the intermediate chain 1 (IC1) and light chain 8 (DYNLL1) of the dynein motor play an important role in the intracytoplasmic retrograde transport of PCV2 along the microtubular track through their interaction with viral Cap protein and tubulin subunits [255]. However, such cytoplasmic movement of PCV2 toward the nucleus is independent of microtubules in monocytic 3D4/31 cells [256]. The disassembly of capsid is influenced by serine proteases under different pH environments in different cells. For example, in monocytic 3D4/31 cells, capsid disassembly requires acidic pH, but in epithelial PK-15 cells the same process requires neutral pH, suggesting the involvement of different types of serine proteases in different cells [257]. The viral genome escapes from the endo-lysosomal degradation pathway and enters the nucleus, where its replication takes place. It is not yet certain whether the PCV2 genome enters the nucleus after disassembly of the capsid or without capsid disintegration. Some experiments suggest that the viral genome enters the nucleus after partial disintegration of the capsid [251], while another proposes that the integrated capsid carries the genome to the nucleus, where it is released without capsid disintegration [256]. Further research is needed to fully understand the intricacies of the PCV2 life cycle.

The ORF1 of the PCV2 genome encodes two differently spliced protein products, Rep and Rep′, which play important roles in DNA replication. Each of these two proteins has N-terminal nuclear localization signals (NLSs), of which NLS1 and NLS2 regulate nuclear import from cytoplasm for the purpose of replication, whereas NLS3 enhances the import mechanism [258]. Both Rep and Rep′ have three conserved RCR motifs, motif-I (FTLN), motif-II (HxQ), and motif-III (YxxK), which are required for ssDNA binding, metal ion binding, and DNA cleavage, respectively, during replication [259]. Motif-III contains a specific tyrosine residue in its catalytic helix, which is essential for site-specific cleavage of dsDNA by a nucleophilic attack, thus generating a 3ʹ-OH primer for strand extension and itself becoming attached to the 5ʹend of the cleavage product [260]. During replication, ssDNA is first converted into a dsDNA intermediate by host enzymes, after which Rep/Rep′ binds with the origin and initiates replication by introduction of a nick using catalytic tyrosine as a nucleophile. The resulting primer is then extended by the host enzymes. Meanwhile, the Rep/Rep′ proteins remain covalently bound to the DNA and terminate the reaction by introduction of a second cleavage reaction within the newly synthesized DNA strand, thus releasing circular ssDNA [258]. The replication of porcine circoviruses is rolling-circle replication (RCR) and probably occurs via a melting pot rather than a cruciform rolling-circle mechanism [260].

The capsid protein (Cap) encoded by ORF2, apart from its structural role in encapsidation, is also crucial both for cytoplasmic transport and replication of PCVs. Cap protein facilitates circovirus trafficking through its interaction with the cytoplasmic dynein IC1 subunit and induction of acetylation of microtubule α-tubulin [261]. Likewise, Cap regulates replication through its interaction with multiple replication-associated proteins such as Rep, pDNAJB6, NPM1, and Hsp70 [262,263,264,265]. The Cap protein has a special arginine-rich N-terminal nuclear localization sequence (position 1–41), which is required for its nuclear import as well as export. After synthesis of the Cap protein in cytoplasm, it is transported into the nucleus by NLS and again exported to the cytoplasm through phosphorylation of NLS [266].

Many viruses use host autophagy for enhancing their replication. PCV2 could also trigger autophagosome formation and promote autophagic flux in PK-15 cells to enhance its replication [267]. The promotion of viral DNA replication via autophagy is catalyzed through repression of mTOR kinase in a cascade of phosphorylated proteins involving AMPK, ERK1/2, and TCS2 [268]. Many host proteins interact with the viral capsid during autophagy-stimulated replication. The interaction of porcine DNAJB6 (pDNAJB6), a major member of Hsp40/DnaJ family with the Cap protein, is thought to promote PCV2 replication by increasing the number of autophagosomes [263]. Oxidative stress might also stimulate PCV2 replication via autophagy [269]. It has been shown that oxidative stress induces autophagy, which in turn promotes PCV2 replication in PK-15 cells through inhibiting the apoptosis pathway [270].

The knowledge on the life cycle of PCV3 is very limited. It is very difficult to isolate and propagate PCV3 in vitro, which makes it difficult to gain scientific information about its infection cycle, pathogenesis, and immunological response. Initially, several attempts to isolate and multiply PCV3 in ST and PK-15 cells failed [23,271]. The first successful isolation of PCV3 in PK-15 cells was performed from tissue samples of perinatal pigs with encephalitis and/or myocarditis, as well as stillborn and mummified fetuses. The replication of isolated PCV3 in PK-15 cells was also confirmed by qPCR, IFA, and RNAscope [109]. At the same time, Oh and Chae successfully isolated and replicated PCV3 using primary porcine kidney cells [272]. Since then, research has begun to shed light on the different aspects of PCV3 biology, including its life cycle. PCV3 enters PK-15 cells by employing calthrin- and dynamin-2-dependent endocytic (CME) pathways. After internalization, PCV3 travels from the endocytic vesicle to the early endosome, and then to the late endosome, and such trafficking depends on Rab5 and Rab7. An acidic environment is required for successful replication of the virus, which might play a role in the process of virus uncoating and genome release [273]. Nucleolar phosphoprotein nucleophosmin-1 (NPM1) is critical for PCV3 replication, as siRNA-mediated knockdown and NPM1 inhibitor treatment markedly reduced the level of PCV3 DNA synthesis, while NPM1 overexpression significantly increased the viral DNA level. The N-terminal NLS domain (1–38) of the Cap protein of PCV3 interacts with NPM-1, and such interaction has a pivotal role in PCV3 replication [274]. More research is needed for further understanding of different entry routes, the role of different proteins in the infection cycle, and the mechanism of PCV3 replication and its regulation.

PCV4 is a newly identified PCV genotype, and detailed information about its life cycle, including its interactions with host cells, way of entry, etc., are currently unknown. Recent studies demonstrated that NPM1 and DEAD-box RNA helicase 21 (DDX21) interact with the N-terminal nucleolar localization signal (NoLS) of Cap protein of PCV4 and play a crucial role in the replication and assembly of viral particles by modulating the translocation of the viral Cap protein from the cytoplasm to the nucleolus of infected cells [275,276].

### 2.9. Prevention and Control

Vaccination is considered to be one of the most effective ways of controlling PCVAD [277]. However, prior to the introduction of vaccines, serum therapy (collection of sera from PMWS-survived, market-age pigs and injection of such hyperimmune serum into young piglets in order to confer protection) has been adopted for the treatment of PMWS, which was later discouraged due to its biosecurity risks [278]. At least five commercial vaccines, one for sow and four for piglets, are available in the global market (Table 3) [124,157]. These vaccine formulations either contain an inactivated whole PCV2a virus (Circovac^®^) or inactivated chimeric PCV1-2a virus (Fostera^TM^ PCV) or subunit proteins of PCV2a capsid (CircoFLEX^®^, Porcilis^®^ PCV, Circumvent^®^ PCV) [279]. All vaccines can reduce clinical sign development, viremia, virus shedding, and pathological lesions, as well as elicit effective humoral and cellular immune responses [280]. In effect, vaccination against PCV2 decreases the mortality rate and improves production parameters such as average daily weight gain (ADWG) [277]. ADWG was found to be significantly higher in a group where both sows and piglets were vaccinated in comparison to groups where either sows or piglets were vaccinated [281]. Vaccination also reduces the detrimental effect of the subclinical infection [282] and number of co-infections [283]. Vaccines elicit neutralizing antibodies, which play an important role in blocking PCV2 replication and preventing lymphoid lesions and clinical disease [284]. The timing of vaccination is important because maternally derived antibodies may interfere with the efficacy of the vaccine. In one study, it was observed that a high level of maternal antibodies at the time of vaccination (3-week-old piglets) was found to interfere with active seroconversion [285]. However, another study demonstrated that the efficacy of the vaccine was not affected by the level of MDA at the time of vaccination (3-week-old piglets) [286]. Thus, further studies are essential to explore the optimal time of vaccination and its relation to vaccine efficacy. Commercial vaccines are based on the PCV2a virus or its capsid protein, but cross-protection against PCV2b and PCV2d has been demonstrated under experimental conditions [284,287,288]. However, one study reported that a vaccine based on PCV2b is more effective at providing protection against PCV2b infection as compared to a PCV2a-based vaccine [289]. Therefore, it is more advantageous to develop a vaccine based on the PCV2b genotype despite the effectivity of PCV2a-based commercial vaccines [290]. Recently, a genotypic shift from PCV2b to PCV2d, along with vaccine failure, has been reported worldwide. The antibody recognition sites in PCV2d are different from those of PCV2a/PCV2b, which might affect the protective efficacy of PCV2a-based vaccines [291]. Previously, it was observed that a live-attenuated chimeric PCV1-2b vaccine could provide equivalent cross-protection against both PCV2b and PCV2d infection [292]. Recently, a PCV2d capsid-based vaccine has also been developed, which was found to effectively reduce PCV2d infection [291]. Therefore, it is essential to develop a vaccine that should have sufficient cross-protectivity or a genotype-based vaccine in order to prevent the severity of PCVADs that are associated with genotypic shifts. Recently, a study aiming to evaluate the efficacy of a virus-like particle (VLP) vaccine based on the Cap protein of PCV2d has demonstrated that rPCV2d VLP has great potential to provide protection against currently prevalent strains of PCV2 (PCV2a, PCV2b, and PCV2d), with an enhanced capability of prevention [293]. To date, the efficacy of different types of vaccine candidates, i.e., DNA vaccines, viral vector vaccines, virus-like particle vaccines, etc., have been tested against PCV2 under experimental conditions [293,294,295,296,297,298,299]. A summary of these studies is presented in Table 4.

At present, no commercial vaccine is available for either PCV3 or PCV4. The protective effect of PCV2 vaccines against PCV3 and PCV4 is also unlikely due to their higher level of genetic and antigenic divergence [300]. Under experimental conditions, PCV3-infected, 3-week-old piglets were vaccinated with a commercially available anti-PCV2 vaccine to assess its impact on PCV3. Despite anti-PCV2 vaccination, PCV3 DNA was detected in the oral fluid and serum of 6- and 8-week-old piglets that were negative for PCV2, thus indicating commercially available anti-PCV2 vaccines are most probably not effective against PCV3 [76]. Recently, soluble sole capsid protein of PCV3 was expressed using an *E. coli* expression system and the purified capsid protein was capable of self-assembly into virus-like particles (VPLs), which has created hope for the development of VPL-based PCV3 vaccines in the future [243]. Similarly, successful expression of PCV4 Cap protein using an *E. coli* expression system was reported and purified Cap self-assembled into VPLs in vitro with high yield. Such PCV4 VLPs could enter PK-15 and 3D4/21 cells. Furthermore, PCV4 VLP-specific mouse serum showed limited or no cross-reaction with PCV2 and PCV3, indicating significant potential of PCV4 VLPs in PCV4-specific vaccine development [301]. An effort to develop soluble chimeric protein consisting of both PCV2d and PCV3 Capsid protein sequences using an *E. coli* expression system for the purpose of vaccine preparation for PCV2d and PCV3 was also undertaken, but this attempt did not lead to successful solubilization [302]. Recently, Tian and his co-workers filed for the patents of a bivalent vaccine for PCV2 and PCV3 [303], as well as for a PCV3 vaccine [304]. However, further studies are needed for the design and development of PCV3 and PCV4 vaccines, and/or multivalent vaccines that provide cross-species protection against all PCVs, as well as their commercialization.

Although vaccination can prevent clinical disease and economic loss by protecting piglets/sows, the complete eradication of the viruses is not yet possible due to the limited protective period of vaccines and re-infection of viruses has been reported when vaccination was stopped [124,305]. Furthermore, vaccination is unable to induce sterilizing immunity and may influence viral evolution in the field [301]. Vaccination is frequently associated with the emergence of vaccine-escaping strains, which ultimately results in vaccine failure [282]. Moreover, vaccination cannot prevent vertical transmission from pregnant sows to fetus [306].

### 2.10. Antiviral Agents

Due to the lack of effective vaccines, there is an increasing demand for alternative approaches to control multifactorial PCVAD. Many studies have aimed to develop antiviral agents using natural compounds derived from medicinal plants and weeds. Some promising findings have emerged from these efforts. Studies have revealed that certain natural compounds exhibit inhibitory effects on PCV2 replication, both in vitro and in vivo. For instance, the polysaccharides of *Sargassum weizhouense*, *Epigallocatechin gallate*, Astragalus, and saponins were found to inhibit PCV2 replication in both cell cultures and mice [307]. In a Chinese study, antiviral activities of twenty natural compounds were evaluated against PCV2 in vitro in Matrine, an alkaloid from *Sophora flavescens* Aiton, and scutellarins, a flavonoid from *Scutellaria* species, were identified as having virucidal activities that interfere with PCV2 replication [308]. Additionally, arctigenin (ACT), a phenylpropanoid dibenzylbutyrolactone lignan extracted from *Arctium lappa* L., has demonstrated a significant inhibitory effect on PCV2 proliferation in PK-15 cells and in mice, which indicates it may be a potential antiviral agent against PCV2 [309]. Furthermore, compounds such as total flavonoids of *Spatholobus suberectus* Dunn (TFSD) have been shown to reduce the ROS-associated pathologies in RAW264.7 cells, indicating their potential therapeutic use [310]. These findings represent a step towards developing effective antiviral agents against PCV2, providing hope for potential alternative treatments to combat the disease in the future.

### 2.11. Improved Managemental Practices

Improvement in the management procedures may also help to prevent or lessen PCV2 infection risks and/or the severity of PCVAD. To reduce the severity of clinical diseases in afflicted farms, Madec’s 20-point plan has been proposed [311]. All-in-all-out practices, disinfection, limiting animal contact, preventing batch mixing, cross-fostering, isolating, or euthanizing sick pigs, maintaining the proper temperature, air-flow, and space within pens, and using appropriate antiparasitic treatments and vaccination are among the recommendations [157,311].

### 2.12. Control of Copathogens

As the co-infecting pathogens play a critical role in the clinical outcome of PCVAD, control of such pathogens could also be effective for reducing the severity of disease. It was observed that vaccination against PRRSV significantly reduces clinical signs of PCVAD, as well as the magnitude of PCV2 viremia [312,313]. However, another study demonstrated that vaccination against PRRSV enhances PCV2 replication, as well as increases PCV2 viremia and clinical signs of PCVAD during later stages of co-infection [314]. Likewise, vaccination against *M. hyopneumoniae* 2–4 weeks prior to PCV2 exposure minimizes the severity of PCV2-associated lesions [161]. However, later studies have found that vaccination against *M. hyopneumoniae* alone did not reduce PCV2-induced lesions [315]. Recently, bivalent (containing PCV2b and *M. hyopneumoniae*) and trivalent vaccines (containing PCV2a/b and *M. hyopneumoniae*) under experimental conditions were tested against dual infections (PCV2d and *M. hyopneumoniae*), which successfully elicited effective protective immunity against concerned pathogens in the form of PCV2-specific neutralizing antibodies and PCV2- and *M. hyopneumoniae*-specific IFN-γ-SC [316,317,318]. Furthermore, trivalent vaccine mixture (3FLEX) was shown to be efficacious against a triple challenge of *M. hyopneumoniae*, PCV2, and PRRSV in pigs [319]. Thus, more research is essential to understand the interaction of different pathogens during co-infection with PCV2 in pigs, host immune responses during such co-infections, and the efficacy of different vaccine strategies and/or combinations for reducing pathological lesions.

Other strategies that can be used to control PCVAD involve the use of disinfectants in buildings and transport vehicles [81]; the use of anti-parasitic drugs such as ivermectin [320]; and addition of anti-oxidant feed additives, conjugated linoleic acid, and spray-dried plasma in the rations fed to nursery-stage piglets [157].

## 3. Conclusions

This review provides a comprehensive overview of porcine circovirus, covering various aspects such as its structure, replication cycle, interactions with host cells, genetic shifts, epidemiology, and preventive measures through vaccines and management practices. The discussion includes the genomic organization and the role of different proteins in the replication and assembly of various PCV genotypes, including PCV1, PCV2, PCV3, and the recently discovered PCV4. Particular emphasis is placed on PCV2 due to its status as the predominant pathogenic virus worldwide. The review highlights the evolutionary trends of PCV2 genotypes, with PCV2a being the dominant genotype until 2003, followed by the emergence of PCV2b as the prevailing worldwide genotype, and more recently, the predominance of PCV2d since 2012. Detailed insights into the pathogenicity of different PCV genotypes and their ability to evade the host’s immune responses are also discussed. Furthermore, the review addresses the global distribution of PCV, its prevalence in domestic pigs and wild boar, and the factors driving the genotype shifts, particularly the transitions from genotype a to b, and subsequently to d. The article provides a summary of recent developments in preventive strategies against PCV infections, such as inactivated vaccines, subunit vaccines, and virus-like particles (VLPs). The efficacy of these newly developed vaccine candidates is thoroughly compared to that of commercial vaccines. The economic implications associated with PCV’s genotypic shifts for the swine industry are acknowledged, underscoring the need for continued research, surveillance, and the implementation of comprehensive strategies to combat PCV effectively. The review emphasizes the importance of further research and development to enhance our understanding of PCV2 and formulate more effective preventive and control measures.

In conclusion, this review highlights the complexity of PCVs and stresses on the different genotypes PCVs to date, and the ongoing efforts required to combat this economically significant pathogen. Through collaborative research and vigilant surveillance, the swine industry can work towards minimizing the impact of PCV and ensuring the health and productivity of pig populations worldwide.

## Figures and Tables

**Figure 1 vaccines-11-01308-f001:**
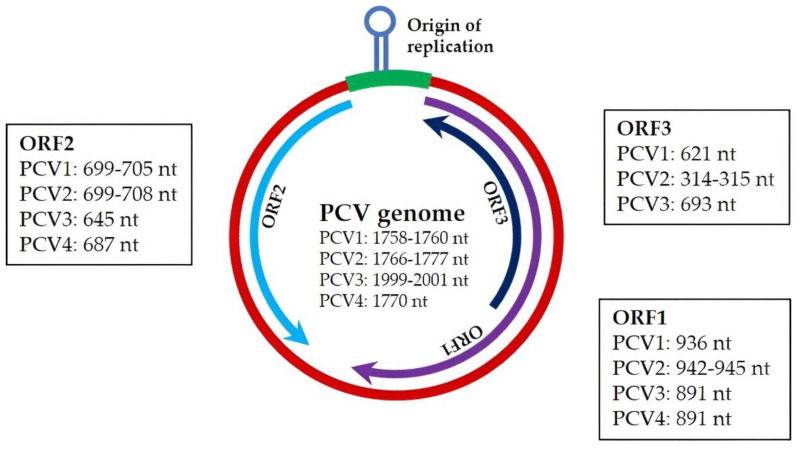
Genome organization of porcine circovirus.

**Figure 2 vaccines-11-01308-f002:**
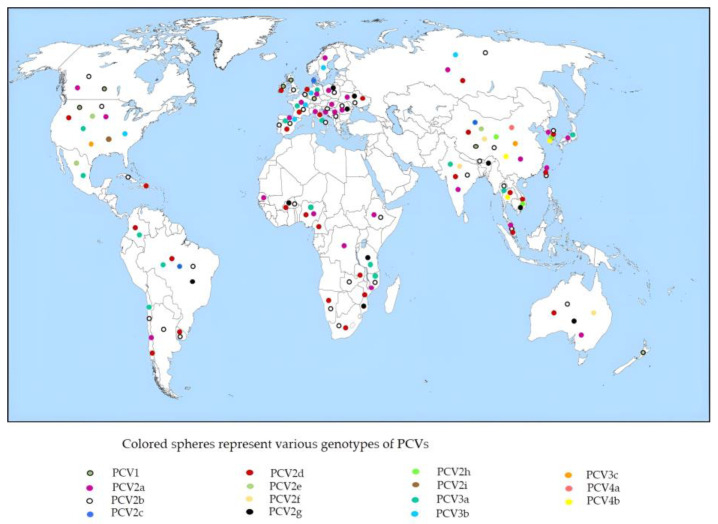
Worldwide occurrence of different species of PCVs.

**Figure 3 vaccines-11-01308-f003:**
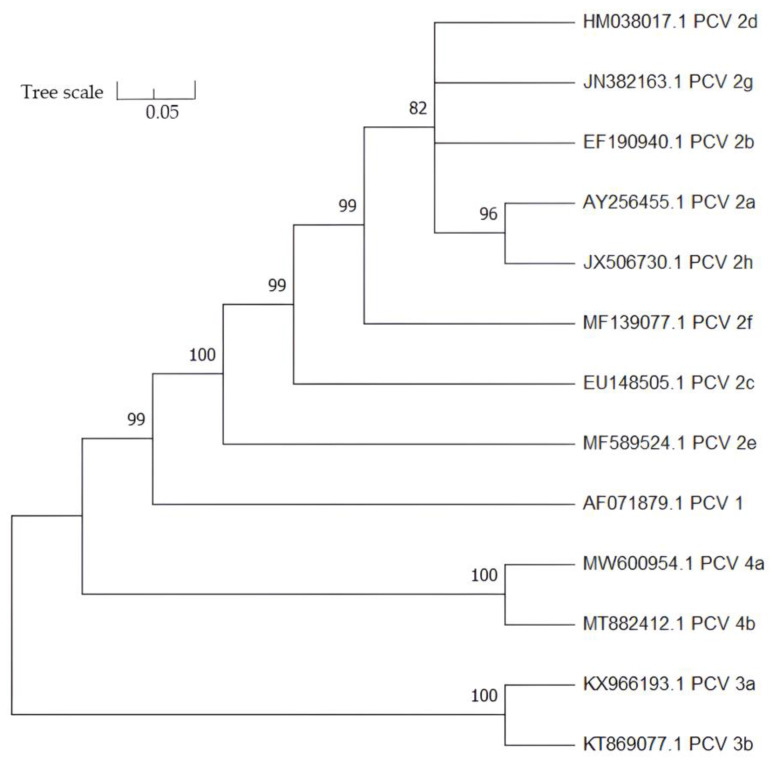
Phylogenetic relationship among four species of porcine circoviruses.

**Figure 4 vaccines-11-01308-f004:**
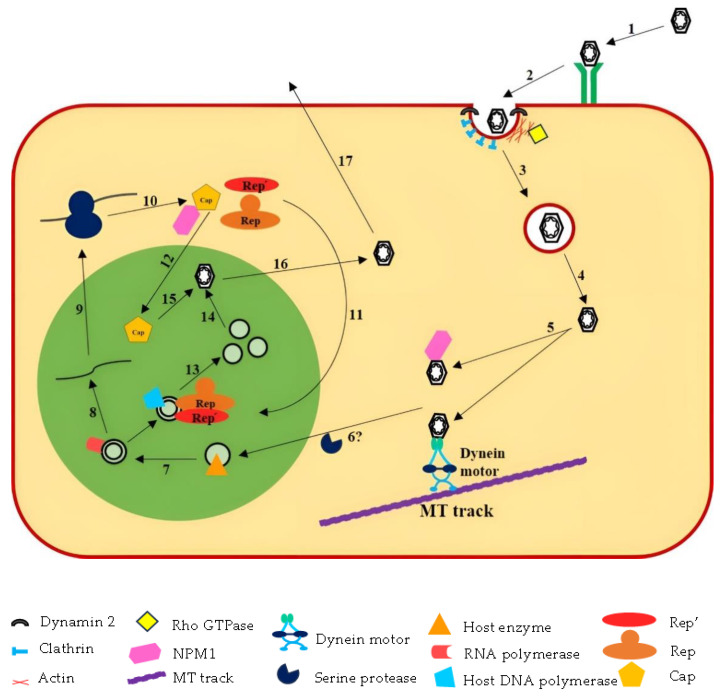
**Life cycle of PCV.** 1 Binding of virus with the cell surface receptor; 2 Entry of the virus via endocytosis/macropinocytosis/actin and Rho-GTPase dependent manner; 3, 4 Transport of the virus through the vesicular trafficking pathway and escape of virus from endosome; 5 Translocation of the virus from the cytoplasm to the nucleus along microtubular track (MT)/without microtubular track. Translocation may involve interaction of virus with NPM1; 6 Disassembly of nucleocapsid by serine protease under acidic or neutral pH and entry of viral genome into the nucleus; 7 Conversion of viral ssDNA into dsDNA by host enzymes; 8, 9, 10 Transcription, translation and synthesis of Cap, Rep, and Rep′ proteins; 11, 12 Transport of Cap (through interaction with NPM1), Rep and Rep′ proteins from cytoplasm into the nucleus; 13 Rolling circular replication of dsDNA by viral Rep, Rep′, and host DNA polymerase; 14, 15 Genome packaging, capsid assembly and generation of new viral particle; 16 Transport of new virus from nucleus to cytoplasm; 17 Export of new virus. Question mark (6) indicates whether or not capsid disassembly occurs prior to the nuclear entry of the virus. Colored shapes represent various proteins and enzymes which are associated with PCV life cycle.

**Table 1 vaccines-11-01308-t001:** Summary of the major ORFs in porcine circoviruses.

Porcine Circovirus	Size (nt)	ORF1		ORF2		ORF3		References
		Protein	Size (aa)	Protein	Size (aa)	Protein	Size (aa)	
PCV1	1758–1760	Rep	312	Cap	230–233	NS	206	[17,18,21]
		Rep′	168					
PCV2	1766–1777	Rep	314	Cap	233–236	NS	104	[17,18,19,21,25]
		Rep′	297					
PCV3	1999–2001	Rep	296–297	Cap	214	Unknown	231	[12,17,19,23,24]
PCV4	1770	Rep	296	Cap	228	-	-	[20]

**Table 2 vaccines-11-01308-t002:** Genomic similarities (%) among porcine circoviruses.

	PCV1–PCV2	PCV1–PCV3	PCV1–PCV4	PCV2–PCV3	PCV2–PCV4	PCV3–PCV4	References
Complete genome (nt)	68.0–76.0	43.5–44.0	50.3–51.6	42.7–48.0	51.5	42.9–45.0	[12,20,21,22,26,27]
Replicase (aa)	86.0	45.5–45.9	48.1–50.7	46.3–48.0	16.2–47.2	48.4–49.7	[20,22,24,26,27]
Capsid (aa)	65.0	24.0–25.2	43.1–44.4	25.9–37.0	12.7–45.0	23.2–24.8	[20,22,23,24,26,27]

**Table 3 vaccines-11-01308-t003:** Details of the currently available commercial vaccines for PCV 2.

Vaccine	Manufacturer	Antigen	Adjuvant	Recommended for	Administration	References
Circovac^®^	Merial (Duluth, GA, USA)	Inactivated PCV2a (whole virus)	Mineral oil	Females of breeding age	2 mL IM 2 doses	[124,157]
Fostera^TM^ PCV	Pfizer (Leipzig, Germany)	Killed PCV1-2a (chimeric virus)	SL-CD aqueous	Piglets (≥3 weeks of age)	2 mL IM 1 dose	
Ingelvac CircoFLEX^®^	Boehringer Ingelheim (Ingelheim am Rhein, Germany)	Cap protein of PCV2a (recombinant)	Carbomer	Piglets (>2 weeks of age)	1 mL IM 1 dose	
Circumvent^®^ PCV	Intervet/SP (Merck, Rahway, NJ, USA)	Cap protein of PCV2a (recombinant)	Microsol Diluvac Forte^®^ (MDF)	Piglets (≥3 weeks of age)	2 mL IM 2 doses	
Porcilis^®^ PCV	Schering-Plough (Merck, Kenilworth NJ, USA)	Cap protein of PCV2a (recombinant)	Mineral oil	Piglets (≥3 weeks of age)	2 mL IM 1/2 dose	

**Table 4 vaccines-11-01308-t004:** Some potential experimental vaccine candidates for PCV 2.

Class	Type of Antigen	Vector Used	Adjuvant	Route of Administration	Effects under Experimental Conditions	Reference
DNA vaccines	Full-length ORF2	pEGFP-N1	Freund’s adjuvant	Intraperitoneal	Immunization of 6-week-old BALB/c mice thrice (15 µg each) at an interval of 14 days provided efficient protection against PCV2 infection through induction of highly specific IgG antibodies and cytokines (IFN-γ and IL-10). Vaccination reduces both viral load and number of microscopic lesions in lymph nodes.	[294]
	Full-length ORF2 (PCV2d)	pVAX1	C3d-P28	Intramuscular	Vaccination of 3-week-old piglets (500 µg each) stimulated both PCV2-specific antibody responses and interferon-γ secreting cells (IFN-γ-SC), reduced viremia and level of genomic DNA and conferred protection against both PCV2b and PCV2d challenge.	[295]
Viral vectored vaccines	Truncated ORF2	Adenovirus (AdEasy^TM^)	CD40L, GMCSF	Intramuscular	Vaccination of 4-week-old pigs induced strong humoral and cell-mediated immune responses and provided better protection than commercial inactivated vaccine (PCV2 SH-strain). Viral load was reduced significantly and no obvious gross and microscopic lesions were observed in lungs and lymph nodes.	[296]
	Full-length ORF2 of PCV2b along with HA Ag of SIV	DS722 (PRRSV)	Not used	Intramuscular	Vaccination of 3-week-old piglets provided good protection against PRRSV but only partial protection against SIV and PCV2b.	[297]
	Truncated Cap	Baculovirus (BacDD)	Not used	Intramuscular	Immunization of 9-week-old SPF pigs (80 µg) induced higher levels of neutralizing antibodies and IFN-γ and significantly reduced viral loads of vaccinated pigs as compared with negative control group.	[298]
Virus-like particles (VLPs)	Full-length ORF2	pET24a (+)	Montanide ISA-201	Intraperotoneal	Vaccination of SPF mice thrice (30 µg each) with rCap VLPs at 2-week intervals induced strong humoral and cellular immune responses as demonstrated by induction of PCV2-specific neutralizing antibodies and secretion of IFN-γ in splenocytes	[299]
	Full-length ORF2 (PCV2d)	pOET1	Not used	Intramuscular	Immunization of 3-week-old piglets induced higher levels of anti-PCV2d IgG and neutralizing antibodies, significantly reduced amount of genomic DNA in blood, saliva, tissues, reduced macroscopic and microscopic lesions in lungs and inguinal lymph nodes and stimulated average daily weight gain in vaccinated group	[293]

## Data Availability

The data (Gene sequences) presented in this study are openly available in GeneBank.

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
