# Peer review of "Revisiting Porcine Circovirus Infection: Recent Insights and Its Significance in the Piggery Sector"

_vaccines, 2023, doi:10.3390/vaccines11081308_

Round 1

Reviewer 1 Report

This is a comprehensive review of the literature relating to most aspects of porcine circovirus, including PCV1, PCV2, PCV3 and PCV4.

A large range of topics, including aspects of the viruses as well as the diseases, is covered. Current and relevant references are used throughout, and the tables and figures provide useful summaries and illustrations. Overall, this is a very thorough and useful review, providing a nice, up to date contribution to the field.

Specific comments:

Infection of wild boar, while mentioned in the abstract and the conclusion, is not addressed in the body of the review.

In table 2, the genetic distances between PCV1 and PCV3 genomes and replicase have not been specified, not for the replicase between PCV3 and PCV4. Could these be calculated by the authors and included?

The English throughout, while good, would benefit from minor editing to improve the style.

In the "clinical signs and pathological lesions" section, at the start of the second paragraph, "symptoms" is mis-spelled.

Author Response

Comments and Suggestions for Authors

This is a comprehensive review of the literature relating to most aspects of porcine circovirus, including PCV1, PCV2, PCV3 and PCV4.

A large range of topics, including aspects of the viruses as well as the diseases, is covered. Current and relevant references are used throughout, and the tables and figures provide useful summaries and illustrations. Overall, this is a very thorough and useful review, providing a nice, up to date contribution to the field.

Specific comments:

  1. Infection of wild boar, while mentioned in the abstract and the conclusion, is not addressed in the body of the review.

Author response: The infection of wild boar with PCVs now has been included in the corrected version throughout the text (Line 129-132, 144-145, 230-233, 264-265, 423-427,443-445)

In table 2, the genetic distances between PCV1 and PCV3 genomes and replicase have not been specified, not for the replicase between PCV3 and PCV4. Could these be calculated by the authors and included?

               Author response: We have included a modified and updated table showing genetic distance of all Porcine circoviruses

Comments on the Quality of English Language

The English throughout, while good, would benefit from minor editing to improve the style.

In the "clinical signs and pathological lesions" section, at the start of the second paragraph, "symptoms" is mis-spelled.

              Author response: The authors attempted to implement editing changes in English, spelling mistakes and grammatical corrections throughout the text, as can be seen in the track changes version. Though minimal unavoidable mistake may still remain.

In the "clinical signs and pathological lesions" section, at the start of the second paragraph, "symptoms" spelling has been corrected.

Reviewer 2 Report

Kartik Samanta et al. provided an updated understanding of the biology, genetic variation, distribution, and preventive strategies concerning Porcine circoviruses and their associated diseases in swine. They comprehensively summarized the biological characteristics of PCV. Overall, this is an interesting review.

1.     The figure resolution is low, please provide high resolution figures.

2.     I found a lot of grammatical errors. Please check English grammar or typography carefully.

3.     As a review, I think it should be attractive, so I suggest that the author should provide a more attractive title.

4.     In the “Genotypes” section: genotypes is very important for prevention of PCV. To give readers a better understanding of the distribution of PCV genotypes, I suggest that the author draw a world map and label PCV genotypes in different countries, at least most of them.

5.     For life cycle of porcine circovirus (figure 2), the authors must provide a complete PCV lifecycle process and should label all relevant proteins rather than each process.

6.     I think the author should provide more information in the “Conclusion” section.

 Please check English grammar or typography carefully.

Author Response

Comments and Suggestions for Authors

Kartik Samanta et al. provided an updated understanding of the biology, genetic variation, distribution, and preventive strategies concerning Porcine circoviruses and their associated diseases in swine. They comprehensively summarized the biological characteristics of PCV. Overall, this is an interesting review. 

  1. The figure resolution is low, please provide high resolution figures.

Author response: In the corrected version, we have included higher resolution images and figure throughout text.

  1. I found a lot of grammatical errors. Please check English grammar or typography carefully.

Author response: Throughout the text, the author attempted to implement editing changes in English, spelling mistakes and grammatical corrections as can be seen in the track changes version. Though minimal unavoidable mistake may still remain.

  1. As a review, I think it should be attractive, so I suggest that the author should provide a more attractive title.

Author response: The title has not been modified

  1. In the “Genotypes” section: genotypes is very important for prevention of PCV. To give readers a better understanding of the distribution of PCV genotypes, I suggest that the author draw a world map and label PCV genotypes in different countries, at least most of them.

Author response: In the corrected version, we have incorporated a higher resolution world map illustrating the occurrence of different genotypes of PCVs worldwide.

  1. For life cycle of porcine circovirus (figure 2), the authors must provide a complete PCV lifecycle process and should label all relevant proteins rather than each process.

      Author response: In the updated version of the article we have included a modified diagram of PCVs replication and life cycle mentioning all the major proteins essential for complete life cycle of PCVs. Role of the all the mentioned proteins in the image were confirmed in various literature. However, we have omitted few protein in the illustration where roles has not been elucidated yet.

  1. I think the author should provide more information in the “Conclusion” section. 

      Author response: In the revised version, the authors have included more comprehensive information in the conclusion section.

Comments on the Quality of English Language

 Please check English grammar or typography carefully.

Author response: Throughout the text, the author attempted to implement editing changes in English, spelling mistakes and grammatical corrections as can be seen in the track changes version. Though minimal unavoidable mistake may still remain.

Reviewer 3 Report

This review paper is written in high quality, only minor concern is to check the abbreviation again, like Line 275  PCV AD, please full name for AD.

Author Response

Comments and Suggestions for Authors

This review paper is written in high quality, only minor concern is to check the abbreviation again, like Line 275  PCV AD, please full name for AD.

 Author response: In the revised version, line 277, the subheading have been changed and full form of  PCVAD  (Porcine circovirus associated disease) is mentioned.

Round 2

Reviewer 2 Report

none

Author Response

We appreciate the rating given by the esteemed reviewer